# DERIVATIVE CAUSAL MODELS: MODELING CAUSALITY AT MIXED SCALES OF OBSERVATION

## ABSTRACT

Causal relations can materialize in many different ways. In their simplest form –typically assumed in classical causal models and discovery approaches–, similar variations of a cause lead to similar variations of an effect. However, this 'smoothness' requires an observation of cause and effect at just the right scales. Unfortunately, this conflicts with records often encountered in the real world, which mix continuous measurements with once-in-a-while observations of sparse events. Compactly modeling the causal effects between (discrete) events and continuous states is difficult to achieve with classical causal models. To ease this situation, we leverage transformations that derive different scales of observables, respectively, to decompose relations and allow for compact causal representations, called *Derivative Causal Models* (DCM). We instantiate them using integral and derivative transforms and demonstrate that the resulting *Differential Causal Models* ($\partial$CM) can be discovered automatically from data.

## 1 INTRODUCTION

There is a growing interest in modeling causal representations in everyday scenarios, such as those commonly encountered in machine learning and applied settings (Schölkopf et al., 2021; Schölkopf, 2022; Lippe et al., 2022; Berrevoets et al., 2023; Runge et al., 2023). While new algorithms are developed to handle an ever-increasing range of scenarios, the underlying causal representations have changed little over the years. As a result, today's causal models often lag behind when it comes to concisely representing complex situations, and it can be quite cumbersome to model certain everyday applications. Classical modeling of even the simplest causal relations, expressed by the simple proposition, "Watering the flower pot will develop a flower", requires knowledge of several (hidden) causal variables, as shown in Fig. 1 ('classical'). Time-dependent relations, such as the lasting humidity state of the soil after the pot is watered, must be explicitly modeled by a human practitioner. While this could be done with lagged causal mechanisms between the event of watering and each time step of the flower size, the resulting representations are rather convoluted and more complex to discover than the connections we want to convey. The resulting representations are rather convoluted and more complex than the simple connection we want to convey. It would be more useful to automatically introduce data transformations between scales of observation, such as sparse events and continuous states, to compactly model such relations. Our approach focuses on relating these different *scales of observation*, as current methods are ignorant of them.

The given example shows possible different qualities of variables: In general, information integration allows us to correlate events that occur at a particular point in time with states that persist even after the initial event has passed. Our approach is particularly useful for modeling systems that relate different scales, such as sparsely occurring information, to persistent states. We anticipate a wide range of applications ranging from modeling 'everyday' causality (e.g. Halpern (2016); Gerstenberg (2022); Zhou et al. (2023)), process analysis (Van Der Aalst, 2012), and climate systems (Runge et al., 2019a; Camps-Valls et al., 2023). We present the advantages of DCM motivated by an everyday introductory example. Furthermore, we show discovery on synthetic time series (Sec. 4.1) and an example of extracting rules of game dynamics (Sec. 4.2).

**Structure and Contributions of the Paper.** Overall, we make the following contributions: (1) We describe how the decomposition of structural equations into data transformations and linear relations can link variables at different scales. (2) We formalize the novel classes of derivative

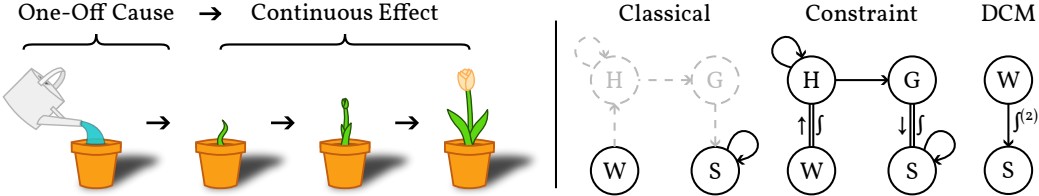

Figure 1: **A one-off cause produces a continuous effect.** Modeling the causal relations of watering a pot leading to a developed flower is inconvenient to represent in 'classical' SCM. First, the one-time event of (W)atering –leading to an increased soil (H)umidity– must be modeled. To discover the causal relation between humidity and plant (S)ize, the derived (G)rowth rate aspect must to be considered. To truly represent such changes, one must consider constraints between the originally observed and its derived quantity (constraint model). Grouping constrained quantities, while showing the difference in transformation levels, yields a representation that is able to compactly convey the simple cause-and-effect relation (DCM; see Sec. 3.2). **(Best viewed in color.)**

causal models (DCM) and differential causal models ($\partial$CM) via a constructive model definition. (3) We propose a compact visualization for $\partial$CM. (4) We propose a greedy score-based algorithm for discovering differential causal models from data. We make our code available at: `https://anonymous.4open.science/r/derivativeCausalModels-E62B/`.

In Sec. 2, we discuss related work and causal models of dynamical systems. Sec. 3 motivates the linear decomposition of structural equations and formalizes an extension of SCM with transformed variables. In Sec. 4 we construct an algorithm to discover $\partial$CM from data. In Sec. 5 we provide a broader perspective on our work and discuss its limitations.

## 2 PRELIMINARIES AND RELATED WORK

In general, we write sets of variables in bold uppercase ($\mathbf{X}$) and their values in lowercase ($\mathbf{x}$). Individual variables and their values are written in normal style ($X, x$). Specific elements of a set are indicated by a subscript index $(X_i)$[1]. Probability distributions over a variable $X$ or a set of variables $\mathbf{X}$ are denoted by $\mathrm{P}_X$ and $\mathrm{P}_\mathbf{X}$, respectively. A detailed list of notations can be found in Appendix A.

SCM provide a framework to formalize a notion of causality via graphical models (Pearl, 2009). The need to search for causal relations beyond the observed variables has already been expressed in other work (Dong et al., 2023). We will start with a standard structural causal model (SCM) and gradually extend the definition in the course of this paper. From a computational point of view, structural equation models (SEM) can be used equivalently to SCM (Halpern, 2000; Spirtes et al., 2000; Rubenstein et al., 2017).

**Structural Causal Models.** A structural causal model is a tuple $\mathcal{M} = (\mathbf{V}, \mathbf{U}, \mathbf{F}, \mathcal{I}, \mathrm{P}_\mathbf{U})$ over variables $\mathbf{X} = \{X_1, \ldots, X_N\}$ taking values in $\boldsymbol{\mathcal{X}} = \otimes_{i \in \{1 \ldots N\}} \mathcal{X}_i$ subject to a strict partial order $<_\mathbf{X}$. $\mathbf{V} = \{X_1, \ldots, X_M\} \subseteq \mathbf{X}, M \leq N$ is the set of endogenous variables. $\mathbf{U} = \mathbf{X} \setminus \mathbf{V} = \{X_{M+1}, \ldots, X_N\}$ is the set of exogenous variables. $\mathbf{F}$ is the set of deterministic structural equations, $V_i := f_i(\mathbf{X}')$, where the parents are $\mathbf{X}' \subseteq \{X_j \in \mathbf{X} \mid X_j <_\mathbf{X} V_i\}$. $\mathcal{I} \subseteq \{\{I_{i,v_i} \mid i \in \mathbf{i}\}_{\mathbf{i} \subseteq \{1, \ldots, M\}} \mid \mathbf{v} \in \boldsymbol{\mathcal{X}} \restriction \mathbf{V}\}$ with $v_i$ for the $i$-th element of $\mathbf{v}$ and $I_{i,v_i}$ indicates an intervention $do(V_i = v_i)$ on an endogenous variable, which replaces the unintervened $f_i$ with a constant assignment $V_i := v_i$. $\mathrm{P}_\mathbf{U}$ is the probability distribution over $\mathbf{U}$.

Each $\mathcal{M}$ induces a directed acyclic graph (DAG) $\mathcal{G} = (\mathbf{X}, \mathcal{E})$ consisting of vertices $\mathbf{X}$ and edges $\mathcal{E}$, where a directed edge from $X_j$ to $X_i$ exists if there are $x_0, x_1 \in \mathcal{X}_j$ such that $f_i(\mathbf{x}', x_0) \neq f_i(\mathbf{x}', x_1)$. For each variable $X_i$ we define $\mathrm{ch}(X_i), \mathrm{pa}(X_i)$ and $\mathrm{an}(X_i)$ as the set of direct children, direct parents and ancestors, respectively, according to $\mathcal{G}$. We define $\mathrm{ch}(\mathbf{X}), \mathrm{pa}(\mathbf{X})$ and $\mathrm{an}(\mathbf{X})$ for sets of variables $\mathbf{X}$, as the union of sets obtained by individual variable evaluations, e.g., $\mathrm{pa}(\mathbf{X}) = \bigcup_{X \in \mathbf{X}} \mathrm{pa}(X)$. In addition, each $\mathcal{M}$ contains an observation distribution $\mathrm{P}_\mathcal{M}$ by propagating $\mathrm{P}_\mathbf{U}$ through the structural

---

[1] When considering sets of variables in this paper, we generally assume that they are indexable via an index function on their unique variable symbol. For notational convenience, we write them as $X_i$.

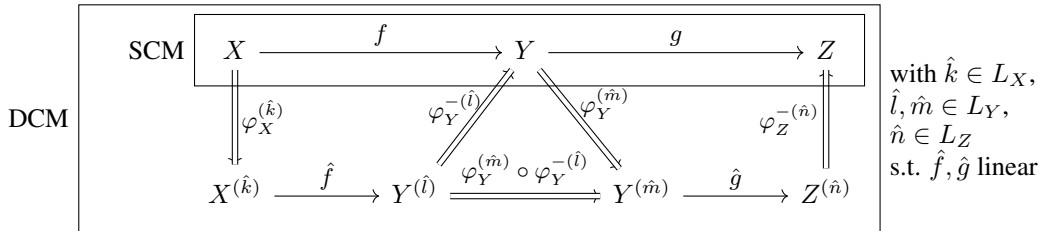

Figure 2: **Decomposition of Structural Equations.** If $X, Y, Z$ are not on the correct scales of observation, respectively, $f$ and $g$ can turn out to be arbitrarily complex. By decomposing the structural equations $f, g$ of a classical SCM (top), we can obtain causal relations containing linear mechanisms $\hat{f}, \hat{g}$ between transformed variables in a DCM.

equations. Each perfect intervention $I$ on a variable $X_i$ replaces a causal dependency $f_i$ with a new probability distribution $P_I$.

**Related Work on Constraint Causal Models and Causal Models with Constraints.** DCM can be understood as a special case of *Constraint Causal Models* (CCM) (Blom et al., 2020) or *Causal Models with Constraints* (CMC) (Beckers et al., 2023), where our focus is on a particular class of constraints that fit suitable data transformations between different scales of observation. From a model-theoretic perspective, DCM are less expressive (but can be more easily recovered as described in this paper) than CMC and we have

$$\mathbf{CCM} \succ \mathbf{CMC} \succ \mathbf{DCM} \succ \mathbf{SCM}.$$

**Related Work on Dynamical Systems and Time Series.** Extensive research has been devoted to the causal study of dynamical systems (Friston et al., 2003; Mooij et al., 2013; Blom et al., 2020; Peters et al., 2022; Löwe et al., 2022) and the general modeling of these (Hyttinen et al., 2012; Mooij et al., 2013; Hansen and Sokol, 2014; Rubenstein et al., 2016; Bongers et al., 2021; Peters et al., 2022). Most often, these works are concerned with solving (e.g. finding unique solutions or converged states of) differential equations (Iwasaki and Simon, 1994; Dash, 2005; Blom et al., 2020; Blom and Mooij, 2023). Such dynamics are common in causal modeling of climate systems (Zscheischler et al., 2020; Camps-Valls et al., 2023; Runge et al., 2023) and in general with time-lagged relationships (Peters et al., 2013; Saggioro et al., 2020; Runge et al., 2019b; Gerhardus et al., 2023; Runge et al., 2023). While our models can be applied to dynamical systems, our goal is to relate variables between different scales rather than to find time-lagged relations. Thus, we do not aim to model feedback systems, but to use differential relations as constraints between related quantities. In general, we focus on identifying instantaneous causal relations within such systems.

## 3 DERIVATIVE CAUSAL MODELS

The goal of our work is to develop a new formalism that is able to integrate observations at different scales within a single causal model. By *scale* we mean the nature of certain variables to exhibit different qualities in observation and thus to be measured by different means and frequencies. Consider again the Watering example in Fig. 1, where the observations span multiple scales. They include one-time events, changes that occur as a quantity over time, and states that measure absolute quantities at a given point in time:

| Event | Change | State |
|---|---|---|
| Soil Watering | Flower Growth | Soil Humidity, Flower Size |

$\longleftarrow$ Scale of Observation $\longrightarrow$

The main difficulty in expressing relationships between scales in classical causal models stems from the fact that these models typically rely on correlational dependencies to infer relationships between variables. However, the one-time signal of the 'Watering' event occurs only at a single point in time, while the resulting flower size is a continuous effect. By simply calculating the correlation between these two observations, they appear to be independent.

**Decomposing Equations.** In theory, arbitrarily complex functional relationships can exist between any two causally related variables. However, such complex mechanisms are generally difficult to understand and difficult to discover. In the simplest case, we can think of causal models that contain only linear mechanisms. We propose to simplify causal models by considering multiple transformed versions of a variable simultaneously, with the goal of reducing the complexity of their structural equations. The transformations that translate between different scales of observation are now considered part of the variables rather than the causal mechanism, and allow expert knowledge to be used to select appropriate transformations that can potentially uncover low-complexity relations. For two variables $X, Y$ connected by a structural equation $Y := f(X)$, there may exist transforms $\varphi_Y, \varphi_X$ with $\varphi_Y \circ Y = \hat{f}(\varphi_X \circ X)$ such that $\hat{f} : X \to Y$ is linear for the right scales of observation. In other words, this would allow the decomposition of a causal mechanism: $Y := f(X) = (\varphi_Y^{-1} \circ \hat{f} \circ \varphi_X)(X)$, where $\varphi_Y^{-1}$ is the inverse of $\varphi_Y$. By separating transforms and causal relations, previously complex structural equations decompose into several simple parts. In cases where certain transforms are shared by multiple variables, the user now only needs to understand the effects of the transform once, rather than having to dissect the multiple, more complex relations. Figure 2 illustrates this decomposition of causal mechanisms and shows how different transformations can be interpreted to induce new 'aspects' of a variable.

Note that $\varphi_X, \varphi_Y^{-1}$ are usually linear operators like convolutions or differential operations, but on the space of possible causal mechanisms, which are not necessarily linear functions.

We begin our formalization of DCM by introducing the notion of transformed variables, which can include multiple scales that are in turn derived from a single observable via transformations $\varphi_i$:

**Definition 1** *Transformed Variable. Given a variable $X_i \in \mathbf{X}$ and a family of bijective transforms $(\varphi_i^{(l)})_{l \in L_i}$, the induced family $X_i^{(L)} := (X_i^{(l)})_{l \in L_i}$ is called a transformed variable, with each $X_i^{(l)}$ determined by $X_i^{(l)} := \varphi_i^{(l)} \circ X_i$.*

We call the individual $X_i^{(l)}$ *aspects* of the transformed variable and sometimes refer to the index $l$ as the *level* of an aspect. We write $\varphi_i^{-(l)}$ to refer to the inverse of $\varphi_i^{(l)}$ and define $\varphi_i^{(0)}$ to be the trivial identity preserving transformation, such that $X_i^{(0)} = \mathbb{1} \circ X_i = \varphi_i^{(0)} \circ X_i = X_i$.

**Admissibility and Interventions.** Since transforms $\varphi_i^{(l)}$ are bijective, they can be thought of as placing constraints between any two aspects $l, l'$, such that $\varphi_i^{-(l)} \circ X_i^{(l)} = \varphi_i^{-(l')} \circ X_i^{(l')}$ which we will denote as $\phi_i^{(l,l')} \in \Phi$ (where $\Phi$ is the set of all constraints). These constraints span the connection to previous *Constraint Causal Models* (CCM) (Blom et al., 2020) and *Causal Models with Constraints* (CMC) (Beckers et al., 2023), where the admissibility of a given configuration of aspects $x_i^{(L)}$ can be validated by testing for $\forall l, l' \in L. \phi_i^{(l,l')}(x_i^{(l)}, x_i^{(l')})$ for all $i = 1, \dots, M$. However, our formalism is more targeted than CCM and CMC, since we require that the relations be defined as functions, rather than allowing more general predicates. In return, we get a constructive process that allows us to compute aspects directly, rather than having to solve for the (possibly implicit) constraints to obtain an admissible solution. Blom et al. (2020) introduce constrained variables, but do not impose explicit constraints on interventions. In such cases, interventions may lead to non-admissible configurations that do no longer satisfy the constraints induced by the transformation (as also discussed in Beckers et al. (2023)). Since we are interested in a modeling approach that guarantees admissibility, we impose certain constraints on our DCM in order to always obtain valid configurations. We will use these assumptions for causal graph discovery in Sec. 4:

**(C1) Only one aspect has parents.** Having parents attached to multiple aspects ($\text{pa}(X_i^{(l)}) \neq \emptyset$ and $\text{pa}(X_i^{(l')}) \neq \emptyset$ for $l \neq l'$) can lead to situations where values from the different parents lead to conflicting configurations, such that $\varphi_i^{-(l)} \circ x_i^{(l)} = x_i^{(0)} \neq x_i^{(0')} = \varphi_i^{-(l')} \circ x_i^{(l')}$. It is unclear how to handle or resolve such situations. At the cost of being more permissive in the set of allowed models, we allow only a single aspect to have a non-empty parent set. Since this is a rather strong constraint on the graph structure, we propose an alternative relaxed version of this constraint in Appendix C.

**(C2) No intervention on multiple aspects.** As discussed above, having multiple 'disagreeing' aspects leads to a situation that may violate the transformation-induced constraints (cf. to Beckers et al. (2023)). As before, the same situation could occur when intervening on multiple aspects

simultaneously. Again, it is unclear which of the possibly conflicting $\varphi_i^{-(l)} \circ x_i^{(l)} = x_i^{(0)} \neq x_i^{(0')} = \varphi_i^{-(l')} \circ x_i^{(l')}$ should be used to infer $x_i^{(0)}$. We therefore work around this problem by defining intervention sets to contain only interventions on a single aspect per transformed variable[2]:

$$\mathcal{I} \subseteq \{\{I_{i,v_i} \mid i \in \mathbf{i}\}_{\mathbf{i} \subseteq \{1,\ldots,M\}} \mid \mathbf{v} \in \boldsymbol{\mathcal{X}} \upharpoonright \mathbf{V}\} \text{ s.t. } do(x_i^{(l)}) \in \mathbf{I} \in \mathcal{I}, \ l \neq l' \implies do(x_i^{(l')}) \notin \mathbf{I}.$$

**(C3) Cutting other aspects' parents on intervention.** Similar to (C2), values induced by interventions may conflict with other aspect values induced by their parents. Therefore, it is necessary to cut the parents of the transformed variable and not only of some aspect.

The constraints introduced for DCM are formal constraints to characterize discoverable DCM rather than strict modeling constraints. In the Flowerpot example, we have four variables: watering, *soil humidity*, *plant growth*, and *plant size*, where *humidity* and *growth* are unobserved. DCM now allows us to model the situation as follows: watering and soil humidity are aspects of a causal entity called *water*, and growth and size are aspects of a second entity called *plant*. A linear causal $humidity \to growth$ relation is modeled. Theoretically, however, we could model a causal relation between any pair of water-plant aspects at the cost of obtaining more complex structural equations between the aspects (cf. Fig. 2, where DCM also includes classical SCM). Note that we can always integrate the aspect transformations of a simple model back into the structural equations. Such marginalization or consolidation approaches have been proposed e.g. by (Rubenstein et al., 2017) and (Willig et al., 2024). However, in the interest of discovering such models, we aim to find the simplest model containing only linear mechanisms.

While we are aware that our constraints may limit the expressiveness of the models, we obtain explicit models whose admissible configurations can be obtained by forward computation within the DCM by the following theorem.

**Theorem 3.1** *Intervened and unintervened evaluations of the SCM under conditions (C1)-(C3) yield admissible variable configurations $\mathbf{x}$ with respect to the transformed variable constraints $\Phi$.*

Having discussed the intuition for conditions (C1)-(C3), we should be able to see the validity of the theorem rather easily. We include a proof in Appendix B for completeness. This also concludes the changes needed to extend SCM in line with the introduction of transformed variables. Thus, we define derivative causal models as follows:

**Definition 2** *Derivative Causal Model (DCM). A derivative causal model $\mathcal{M}^{\boldsymbol{\varphi}} = (\mathbf{V}^{\boldsymbol{\varphi}}, \mathbf{U}^{\boldsymbol{\varphi}}, \mathbf{F}^{\boldsymbol{\varphi}}, \Phi^{\boldsymbol{\varphi}}, \mathcal{I}^{\boldsymbol{\varphi}}, \mathrm{P}_{\mathbf{U}}^{\boldsymbol{\varphi}})$ is an extension of an SCM $\mathcal{M} = (\mathbf{V}, \mathbf{U}, \mathbf{F}, \mathcal{I}, \mathrm{P}_{\mathbf{U}})$, under constraints (C1)-(C3), induced by a set $\boldsymbol{\varphi} = \{(\varphi_i^{(l)})_{l \in L_i} \mid i = 1, \ldots, N\}$ of families of bijective transformations and a specific set of aspect indices $\mathbf{l}^{\star} \in \otimes_{\{j|X_j \in \mathbf{X}'\}} L_j$ with*

$$\mathbf{V}^{\boldsymbol{\varphi}} := \{X_i^{(l)} \mid i = 1, \ldots, M; \ l \in L_i\}$$

$$\mathbf{U}^{\boldsymbol{\varphi}} := \{X_i^{(l)} \mid i = M+1, \ldots, N; \ l \in L_i\}$$

$$\mathbf{F}^{\boldsymbol{\varphi}} := \{V_i^{(l_i^{\star})} := f(\mathbf{X}') \mid i = 1, \ldots, M; \ l_i^{\star} \in \mathbf{l}^{\star}\}$$

$$\Phi^{\boldsymbol{\varphi}} := \{V_i^{(l)} = \varphi_i^{(l)} \circ \varphi_i^{-(l_i^{\star})} \circ X_i^{(l_i^{\star})} \mid i = 1, \ldots, M\}$$

$$\mathcal{I}^{\boldsymbol{\varphi}} := \{\{do(V_i^{(l)} := v_i^{(l)}) \mid i \in \mathbf{i}\}_{\mathbf{i} \subseteq \{1,\ldots,M\}} \mid \mathbf{v} \in \boldsymbol{\mathcal{X}} \upharpoonright \mathbf{V}; \ \mathbf{l} \in L_1 \otimes \cdots \otimes L_M\}$$

$\mathbf{l}^{\star}$ serves as an indicator set for which aspect of a transformed variable is determined by its parents. All other $l \neq l^{\star}$ are then determined by the constraints $\Phi^{\boldsymbol{\varphi}}$ between aspects. (When being free to choose, it may be advisable to choose $\mathbf{l}^{\star} = \mathbf{0}$ in order to have child aspects appear in structural equations with index 0.)

Note that the definition of $\mathbf{X}'$, constrained by the partial order $<_{\mathbf{X}}$, remains unchanged, while the semantics of $<_{\mathbf{X}}$ changes slightly. Due to the constraints $\Phi^{\boldsymbol{\varphi}}$ induced by the transformations, all aspects are required to be on the same 'level' as the original observable. That is: $X_i <_{\mathbf{X}} X_j \iff X_i^{(l)} <_{\mathbf{X}} X_j^{(l')}$ for all $i, j; l, l'$. This completes our formalization of DCM.

---

[2]Technically, multiple interventions could be allowed, as long as they all result in the same $x_i^{(0)}$. Due to constraints (C1)-(C3), this is equivalent to choosing any single intervention from the set of agreeing interventions.

Consider furthermore, that more than one aspect, e.g. the pair $X_i^{(l)}, X_i^{(l')}$, could be governed independently via different structural equations: $X_i^{(l)} := f(\mathbf{X}')$ and $X_i^{(l')} := g(\mathbf{X}')$. We could then transform $X_i^{(l')}$ back to $X_i^{(l)}$ which then yields $X_i^{(l)} := \varphi^{(l)} \circ \varphi^{-(l')} \circ g(\mathbf{X}')$. However, this means that $f(\mathbf{X}') = \varphi^{(l)} \circ \varphi^{-(l')} \circ g(\mathbf{X}')$. Since $f$ already fully determines $X_i^{(l)}$, we would have to require that $g$ (via the transformation $\varphi^{(l)} \circ \varphi^{-(l')}$) follows $f$ and computes the exact same value for $X_i^{(l)}$. While we could allow for such scenarios, $g$ will not add any more information than is already given by $f$. Even more so, the probability of $f$ and $g$ disagreeing on particular values of $X_i^{(l)}$ is high if we consider sampling random models from the unconstrained model space. In these cases, the probability of the data collapsing to zero makes the model inadmissible, which is what we avoid with our constraints.

### 3.1 Differential Transforms in Causal Models

Some choices of transforms may be more convenient than others. In the following, we will focus on the particularly useful class of differential transforms. Differential quantities of a variable can be related via their anti-derivatives (e.g., Bongers et al. (2018), Blom et al. (2020, Eq. 2)):

$$X(\dot{x}_{0..t}; c) = c + \int_0^t \dot{X}_s \, ds \tag{1}$$

The equation 1 gives rise to different aspects of a causal variable. For each variable, we can consider the observed quantity $X$, its n-th derivatives $(\dot{X}, \ddot{X}, \dots)$, and its anti-derivatives $(\int X, \int\int X, \dots)$. Given the starting conditions $\mathbf{c}$, we can infer the integral representations and/or compute the derivatives of a variable. Thus, each variable induces its *derivative family* by taking derivatives and/or anti-derivatives (wherever they exist):

**Definition 3** *Differential Transform. The differential transform operator is recursively defined by:*

$$s(k, l, \mathbf{c}) = \begin{cases} (c_{k+l} + \int) \circ s(k, l-1, \mathbf{c}) & \text{if } l > 0 \\ (d/dt) \circ s(k, l+1, \mathbf{c}) & \text{if } l < 0 \\ \mathbb{1} & \text{if } l = 0 \end{cases} \tag{2}$$

The operator $s(k, l, \mathbf{c})$ with $k, l \in \mathbb{Z}$ induces the set of differential transforms $\varphi_i^{(l)} := s(0, l, \mathbf{c}_{X_i})$, where $\mathbf{c}_{X_i}$ is the starting vector corresponding to $X_i$. $k$ is the absolute level of the aspect to which we apply the transformation, and $l$ can be used to perform relative level shifts. To avoid having to talk about transformed variables with differential transforms, we define differential causal variables:

**Definition 4** *Differential Causal Variable. A differential causal variable is a transformed variable induced by the family of differential transforms $X_i^{(l)} := s(0, l, \mathbf{c}_{X_i}) \circ X_i^{(0)}$.*

We can observe some regularities following the two previous definitions. In particular, note that for differential causal variables the relation $\varphi_i^{-(l)} = \varphi_i^{(-l)}$ holds, since $s^{-1}(0, l, \mathbf{c}_{X_i}) = s(0, -l, \mathbf{c}_{X_i})$. Additionally, we can relate any two $X_i^{(k)}, X_i^{(k+l)}$ to each other by their index level difference $X_i^{(k+l)} = s(k, k+l-k, \mathbf{c}_{X_i}) \circ X_i^{(k)} = s(k, l, \mathbf{c}_{X_i}) \circ X_i^{(k)}$. Thus, knowing the relative level difference between aspects is sufficient to translate between them. This is especially useful when we want to discover new relations, but do not know the exact level at which an observation is made. Thus, all aspects can be transformed into each other by applying relative shift transforms. In short, we make fewer assumptions about our observations and directly create transformed variables in the range $[l-n, l+n]$ up to a user-defined $n$, without worrying about the absolute scale $l$ of an observable. Finally, we define differential causal models as a special class of derivative causal models:

**Definition 5** *Differential Causal Model ($\partial$CM). A differential causal model is a derivative causal model in which all transforms are differential transforms.*

An often considered special case of $\partial$CM is that of (discrete) time series, which appears in a variety of scenarios such as climate science, finance, or reinforcement learning (Runge et al., 2023; Hasan et al.,

2023; Geweke, 1984; Bareinboim et al., 2015; Sutton and Barto, 2018). Even if there are no explicit observations of derivatives or integrals are available in the data, we can induce these quantities by computing finite differences. To make the transformation bijective, as required by Def. 1, we need to know a starting vector $\mathbf{c}$. For the later application to discrete time series, we assume that the values of the previous time step $t - 1$ are to be observed. In other scenarios, such as measuring the position of a car, its derivative 'speed' and integral 'distance traveled' values may be read from instruments and provided within the data. While previous work considers the time aspect of the data as a means to infer delayed causal relationships, we use the time dimension exclusively to obtain integral and/or derivative values.

### 3.2 Compact Derivative Causal Model (DCM) Representation

Since we wanted to present a simplified causal representation at the beginning, we will now briefly discuss a possible compact representation for graphs containing transformed variables. The notion of variables linked by differential transforms imposes constraints on aspects. As a consequence, individually transformed aspects do not stand alone, but can be grouped together. Depending on the use case, this reduces the number of nodes and makes 'higher level' qualitative relationships more visible. Transformed variables (and thus differential causal variables) can be thought of as forming cluster DAGS (Anand et al., 2022), causal composition variables (Willig et al., 2024) or coarsened graphs (Wahl et al., 2023) (all, mutas mudandis the replacement of directed structural equations by constraint relations). These and other previous works (Zhu et al., 2024; Kekić et al., 2023) generally do not consider 'symmetric' undirected constraints, but assume directed structural equations between variables as to derive identification and independence properties between clusters. Our formulation was specifically chosen to ensure that the coarsened graph (Wahl et al., 2023) remains acyclic.

Since we want the DCM graph to convey the qualitative nature of causal relations, we use the relative nature of $\partial$CM transforms to indicate the offset in transformation levels between two variables. For example, in the 'constraint' representation of Fig. 1, we see that the causal effect passes through two integral transforms. In our example, this is the result of the initial event-to-state aggregation $(C -^{\int}\rightarrow W)$, the following equal-scale causal relation $(W \rightarrow G)$, and a final integral transformation $(G -^{\int}\rightarrow S)$. With some imprecision, we pull the integrals together along the chain to convey the qualitative –double integrating– nature of the relation. In general, we expect most simple Event $\rightarrow$ State relations to be of the form $X -^{\int^{(2)}}\rightarrow Y$. Alternatively, a slightly less convenient way to write them is to represent them in terms of derivative transforms: $G -^{(dx/dt)^{(-2)}}\rightarrow S$. Note that whenever $l$ and $m$ have different signs, it may be advisable to list them separately as $G -^{(\int^{(+l)}, \int^{(-m)})}\rightarrow S$ to avoid confusion due to cancellation effects.

## 4 Discovering Differential Causal Models from Data

The previous sections were primarily concerned with the formalization of derivative and differential causal models for modeling systems and processes. We recognize that –in the field of causality– there is a great interest in the (automated) discovery of causal structures from data. In the following, we develop the Greedy Derivative Model Search (GDS) algorithm for discovering differential causal models using a score-based approach.

Navigating the class of Markov-equivalent models, as in the GES algorithm (Chickering, 2002), is challenging for DCM because our constraints impose additional restrictions on the set of edges that can be added at each step. In particular, we are often prevented from adding or inverting certain edges as would be done in classical GES. We take inspiration from the GLOBE algorithm of Mian et al. (2021), which exploits the minimal representation length (Janzing and Schölkopf, 2010) to still perform a graph search over the class of SCM. We also modify the standard Bayesian Information Criterion (BIC; Schwarz (1978)) evaluation to significantly improve performance in our setting. As a major benefit, we can verify that our new DCM constraints can be applied during the edge selection phase. Pseudocode for the algorithm can be found in Appendix D. A Python implementation of our algorithm and the evaluation scripts used can be found in the code repository: https://anonymous.4open.science/r/derivativeCausalModels-E62B/.

| GDS | Precision (↑) | Recall (↑) | SID (↓) |
|---|---|---|---|
| $\alpha = 0.01$ | **63.47** (20.44) | 60.21 (22.16) | 4.95 (3.82) |
| $\alpha = 0.1$ | 62.87 (20.34) | 59.50 (22.48) | 5.00 (3.84) |
| $\alpha = 1.0$ | 61.03 (24.78) | 49.19 (21.07) | 6.10 (3.98) |
| $\alpha = 10.0$ | 62.66 (30.54) | 28.88 (15.61) | 8.85 (3.75) |
| $\alpha = 100.0$ | 61.16 (35.28) | 26.21 (17.84) | 8.80 (3.85) |
| linear regr. | 60.74% (19.92) | **60.59%** (24.73) | **4.85** (3.90) |
| no poly. fit. | 25.68% (15.08) | 38.40% (24.56) | 9.80 (4.01) |
| GES (std) | 28.92% (14.19) | 36.30% (34.63) | 9.76 (5.50) |
| GES (pf) | 21.24% (13.16) | 33.36% (29.42) | 10.76 (3.55) |

Figure 3: **Graph Constraints and Evaluation of Greedy Differential Search. (Left)** The figure visualizes forbidden edges induced by the conditions of Sec. 4.1 (red), in a graph with existing edges (green). (Additional non-candidate edges are not shown for clarity.) Edge (1) creates a cycle within variables of the coarsened graph (dashed rectangles). Edge (2) would connect aspects of the same transformed variable. Edge (3) adds a parent to a new aspect, while another aspect already contains parents. **(Right)** Evaluation of the GDS algorithm for different settings of $\alpha$. We compare to GES with standard (std) and our polynomial fitting (pf) loss. (Best results in **bold**; second place in underlined). The mean (and standard deviation) is given in parentheses. SID (lower is better ↓) is computed over the coarsened graph. Small sparsity regularization generally improves the accuracy. More results can be found in Appendix D.1. **(Best viewed in color.)**

## 4.1 GREEDY SEARCH FOR DISCOVERING DIFFERENTIAL CAUSAL MODELS

Like GLOBE, our GDS algorithm consists of a forward and a backward phase that are repeated many times until convergence. In a forward phase, the algorithm iteratively selects new edges to add until no more edges can be added without increasing the score. The score of each edge addition is the difference between the unaltered graph and the graph with the candidate edge added. Since this is a greedy heuristic, it runs the risk of getting stuck in local optima. For example, a node initially explained by a single edge with a high score might be better explained by two separate weaker edges added later. To eliminate such edges, an additional backward phase is performed, removing individual edges from the parents and testing whether the score still improves (Mian et al., 2021). We repeat this process up to 5 times or until convergence.

In a first step we augment the data with transformed aspects of a variable. As discussed earlier, we recursively compute $X_t^{(l)}$ from $X_t^{(l\pm1)}$ and require that the starting values of the previous time step $X_{t-1}^{(l)}$ are known. In the discrete time step setting we compute integrals by simple summation $X_t^{(l+1)} := X_{t-1}^{(l+1)} + X_t^{(l)}$ and derivatives by a finite differences $X_t^{(l-1)} := X_{t+1}^{(l+1)} + X_t^{(l)}$. Obviously, this augmentation requires that the number of transformations considered, $L$, be finite. For our DCM experiments, we consider a setting with one addition, derivative, and integral aspect per variable, resulting in an effective tripling of variables in our case.

To make the algorithm follow the DCM constraints, we impose several constraints on the selection of possible candidate edges. Specifically, we impose the following constraints (see Fig. 3 (left)):

**(1) Acyclicity.** As with classical SCM, we must ensure that the coarsened graph remains acyclic. Since the transformation constraints between different aspects are not reflected in the data, we obtain the coarsened graph in each step and exclude edges that would lead to cycles.

**(2) In-constraint Edges.** Furthermore, we do not allow edges between aspects within a constraint variable, since these relations are already determined by the transformations (and thus would induce self-cycles within the global graph). This follows directly from the irreflexivity of the strict partial order $<_\mathbf{X}$ over the coarsened graph, such that $\forall l, l' \in L_i. X_i^{(l)} \not<_\mathbf{X} X_i^{(l')} \Rightarrow (X_i^{(l)}, X_i^{(l')}) \in \mathcal{E}$.

**(3) Multiple Child Aspects.** This constraint is imposed to prevent multiple (possibly conflicting) signals from being present on the aspects, as required by (C1). Only a single aspect within each transformed variable can contain parents. Once an aspect within a transformed variable receives a parent edge, all other parents are prohibited from receiving additional incoming edges. This is the most restrictive constraint in terms of allowed graph structures, and we hope to relax it in the future. It

follows from the uniqueness of a single aspect with parents within each transformed variable, implied by $l^\star$: $(l_i^\star \in L_i) \Rightarrow (\forall l_i^\star, m_i^\star \in L_i. ((\mathrm{pa}(X_i^{(l_i^\star)}) \neq \emptyset) \Rightarrow (\mathrm{pa}(X_i^{(m_i^\star)}) = \emptyset \vee l_i^\star = m_i^\star)))$

**Bayesian Information Criterion.** We use the BIC to estimate the cost of adding a new edge to a model. Specifically, we compute the common formulation $BIC = n * \ln(\hat{\sigma}^2) + k * \ln(n)$ (Priestley, 1981), where $\hat{\sigma}^2$ are the prediction residuals after linear regression (and polynomial fitting; see below), $k$ is the number of model parameters, and $n$ is the number of samples. As a first step, we fit the parental weights using linear regression. In addition, when fitting the parents, we apply the LASSO regularization (Tibshirani, 1996) to zero out the weights of non-parent candidates in cases where ordinary least squares would still assign some average weight. LASSO introduces an $\alpha$ parameter that balances the goodness of fit with the sparsity of the weight vector via L1 regularization. Whenever a previously included parent weight regresses to zero, it is removed from the set of parent candidates.

**Polynomial Fitting.** Because the data have aspects at different scales, the residual of a variable (especially integral aspects) is most likely to exhibit non-Gaussian behavior. Computing the variance of an integral aspect –that is, over the residuals after linearly fitting the parents– as is done in the standard BIC calculation, provides a strong incentive for the algorithm to predict integral aspects first in order to reduce excessive variance. To account for the potentially nonlinear behavior of the residuals, we fit multiple polynomials to a given degree and only then compute the remaining variance (see Alg. 4). The parameters needed to fit the polynomial are counted in the number of model parameters $k$. The polynomial fit with the lowest BIC is then selected for further comparison with other edges. Our experiments confirm our modeling choice, as we observe a *severe drop in performance* when no polynomial fit is performed on the linearly fitted residuals (see Table 3).

**Evaluation.** We evaluate our algorithm on random graphs with 6 nodes and a default edge probability of 50%. (See Appendix D for extensive evaluations varying the number of nodes and edge percentages.) We generate one derivative and one integral aspect per variable and sample 10 initial configurations. Each configuration is run for 50 time steps. Exact parameter details are given in Appendix D.1. We measure edge recall and precision (higher is better ↑), as well as structural intervention distance (Peters and Bühlmann, 2015, SID) (and structural Hamming distance (Acid and de Campos, 2003; Tsamardinos et al., 2006, SHD) (lower is better ↓) in Appendix D) on the coarsened graph. (We are not aware of any variant that is able to measure SID in the presence of transformed variables). Full experimental details are given in Appendix D.1.

To the best of our knowledge, no causal discovery algorithm has been proposed to discover instantaneous effects under transformation constraints. Similar baselines, such as GES or GLOBE, do not exploit constraints and do not guarantee that they lead to valid DCM. Nevertheless, GES should be able to identify the causal relationships when given the individual aspects. We therefore report the performance for GES as a baseline algorithm, unaware of the constraints that may be present in the system. (Exact evaluation details are given in Appendix D.1). We evaluate each configuration over 20 seeds and report the results in Figure 3. We vary the $\alpha$ of the LASSO regression, compare it to ordinary least squares regression, and evaluate our algorithm without polynomial fitting of the residuals. We find that the small LASSO regularization with $\alpha = 0.01$ outperforms ordinary least squares regression in terms of precision, while linear regression is slightly better in terms of recall and SID score. As expected, fitting without polynomial fitting of the residuals significantly reduces the performance. Overall, we observe a high standard deviation, which we attribute to the aforementioned problem of a difficult to navigate search space. Selecting the wrong edges (or edge directions) at the beginning of the algorithm makes recovery difficult.

**Application to a Real-World Maize Watering Example.** Finally, the discovery algorithms are applied to the publicly available real-world data set 'USDA-ARS Colorado Maize Water Productivity Dataset'. (See Sec. D.1.1 in the appendix for more details). Although no ground truth graph exists, the dataset is similar to the plant growth example from the introduction, containing sparsely occurring 'watering' factors such as 'irrigation' and 'precipitation', and 'plant size' factors such as 'root depth' and 'canopy cover'. In fact, the GDS algorithm is able to correctly identify the predicted integral relationships, in particular, the causal relationships 'Irrigation → Root Depth' and 'Irrigation → Canopy Cover' are correctly inferred. In contrast, GES is unable to detect any integral or derivative relations. The recovered causal graphs, together with a brief discussion of the results are respectively presented in Figure. 5 and Sec. D.1.1 in the Appendix.

| Movement | Interactions |
|---|---|
| moveLeft $-\!\!\int\!\!\rightarrow$ posX | coll.Key $\overset{-\int\rightarrow}{=\!=\!=}$ hasKey |
| moveRight $-\!\!\int\!\!\rightarrow$ posX | hasKey $\rightarrow$ doorOpen |
| moveUp $-\!\!\int\!\!\rightarrow$ posY | coll.Gem $-\!\!\int\!\!\rightarrow$ score |
| moveDown $-\!\!\int\!\!\rightarrow$ posY | coll.Door $\rightarrow$ finished |

Figure 4: **Capturing Game Dynamics. (Left)** Individual frames of a simple game walk through. **(Right)** Game dynamics described by $\partial$CM relations. Possibly *continuous* Input $-\!\!\int\!\!\rightarrow$ State relations connect the player inputs and avatar position, while Event $-\!\!\int\!\!\rightarrow$ State interactions describe *discrete* states related to game logic. A full description is provided in Appendix E. **(Best viewed in color.)**

## 4.2 Capturing Game Dynamics

As a last example, we present a brief illustration to demonstrate the usage of $\partial$CM for modelling game dynamics. This could be useful to guide agent training or reason about the root causes of agent actions (Schölkopf et al., 2021; Brehmer et al., 2022; Lachapelle et al., 2022; Weichwald et al., 2022; Kumar et al., 2024; Talon et al., 2024). Consider the game environment depicted in Figure 4. In this environment, a variety of observables among different scales can be observed in every frame. These include events, such as moveLeft, collidesKey, or states such as doorOpen, score. In a first step we perform a *semantic grouping* where observables of different scales of the same object (e.g. collideKey and hasKey) are grouped into a single transformed variable. (Indicated in the figure by $A =^{-\int}\!\!= B$). Using $\partial$CM spares us from having to model the explicit constraints over the different variables every time such a situation arises and, compared to classical SCM, allows us to handle the relations between the two aspects in a more uniform manner. In addition to the apparent relation between movement and position, we are also able to model integral rules in the environment, such as collectGem $-\!\!\int\!\!\rightarrow$ score. Note that even in the case of not observing hasKey explicitly, $\partial$CM would still be capable of deriving the collectKey $-\!\!\int\!\!\rightarrow$ doorOpen causal relation. While these relations may be modelled using other frameworks (e.g. [extended] summary causal graphs, (Assaad et al., 2022b;a)), the use of $\partial$CM facilitates a more effective disentanglement of dynamics and interaction rules as qualitative relations in the environment. In Appendix E, we provide a brief example on how to leverage the presence of derivative aspects in $\partial$CM to reason backwards in time and infer information about previous states of the environment.

## 5 Conclusion

The observation of causal relations between single events and their continuous effects is a common occurrence, particularly within everyday situations. The ability to concisely represent such causal connections is a challenging, yet key requirement for the correct application of causal models. To the best of our knowledge, we are first to conceptualize the notion of scales in DCM and $\partial$CM. The approach enables practitioners to model real-world scenarios more concisely, thereby allowing them to reason more effectively on the causal relations across different scales. In particular, the seamless embedding of integral aspects is if great importance, as they *accumulate information* about the past states into current observations. This property may be employed to reason backward in time (e.g. performing Taylor approximations with the help of derivative aspects) to reconstruct process sequences or trace back root causes using observations alone. To discover DCM and $\partial$CM automatically from data, we propose the Greedy Derivative Model Search algorithm, which is able to recover DCM from data under the presence of aspect transformations.

**Limitations and Broader Impact.** The examples provided considered the instantaneous effects between variables. In other words, our experiments did not consider time-lagged relations between variables, apart from the aggregational dynamics of integral quantities. We expect future work to also incorporate time dependencies, including the incorporation of lagged connections. Although our definition of DCM has several useful properties, it also imposes certain restrictions on the graph structures that can be modelled. We have relaxed these constraints in a discussion in Appendix C and hope to develop more general notions of DCM in the future. In general, we expect our newly proposed DCM and $\partial$CM to enable practitioners to model real-world problems more concisely and identify qualitative relations more easily.

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

SUPPLEMENTARY MATERIAL: "DERIVATIVE CAUSAL MODELS: MODELING CAUSALITY AT MIXED SCALES OF OBSERVATION"

## A  MATHEMATICAL SYMBOLS AND NOTATION

The following table contains mathematical notation and functions used throughout the paper.

| Notation | Meaning |
|---|---|
| $X$; $\mathbf{X}$ | An (indexed set of) variable(s). |
| $x$; $\mathbf{x}$ | Value(s) of $X$; $\mathbf{X}$. |
| $\mathbf{X}_i$ | The i-th variable of $\mathbf{X}$. |
| $\mathbf{X_S}$ | The subset $\{\mathbf{X}_i : i \in \mathbf{S}\}$ of $\mathbf{X}$. |
| $\mathrm{P_X}$ | A probability distribution over variables $\mathbf{X}$. |
| $\mathcal{X} \upharpoonright \mathbf{V}$ | Restriction of the domain of $\mathcal{X}$ to the subset of the domain over $\mathbf{V}$. |
| $f \circ g$ | Function composition, $(f \circ g)(x) = f(g(x))$. |
| $\otimes_{X_i \in \mathbf{X}} \mathcal{X}_i$ | N-ary Cartesian product over the domain of $\mathbf{X}$. |
| $\lVert \cdot \rVert_2$ | $l^2$ vector norm. |
| $\mathcal{U}(a, b)$ | Uniform Distribution. |
| $\mathcal{N}(\mu, \sigma^2)$ | Normal Distribution. |

## B  PROOF OF THEOREM 3.1: ADMISSIBILITY OF VARIABLE CONFIGURATIONS IN DCM

In this section we prove Theorem 3.1 of the main paper. To do so, we need to show that the constraints induced by the transforms (Sec. 3) are always fulfilled during normal evaluation and under intervention. The main point is to show that no two conflicting configurations are induced by parents or interventions at any point in time. As constraints are placed per transformed variable it is sufficient to prove the validity of constraints for arbitrary transformed variables to prove the validity of Theorem 3.1 for the whole SCM:

*Unintervened Evaluation:* Assume that no intervention is present at any $X_i^{(l)} \in X_i^{(L)}$. By constraint (C1) as realized in Def. 2 we require only $X_i^{(l^\star)}$ to have parents. (Note that this also excludes any other potential intra-aspect causal relations in $X_i^{(L)}$). By Def. 2, all other $X_i^{(l)}, l \neq l^\star$ are defined via $X_i^{(l)} = (\varphi_i^{(l)} \circ \varphi_i^{-(l^\star)})(X_i^{(l^\star)})$, which is the exact definition of constraints $\phi$ in Sec. 3 (as included in Def. 2). Finally, we need to show that no other competing source of information (via other aspects containing parents or interventions) breaks the constraints. By constraint (C1) no other aspects contains parents and no intervention is present by assumption. Therefore, no other –potentially conflicting– signal influences $X_i^{(l)}$ and the values for all aspects are uniquely computed for the unintervened case.

*Intervened Evaluation:* Assume that an intervention $do(X_i^{(l)} := x_i^{(l)})$ is present at some $X_i^{(l)} \in X_i^{(L)}$. By Def. 2, all other $X_i^{(l')}, l' \neq l$ are defined by $X_i^{(l')} = (\phi_i^{(l')} \circ \phi_i^{-(l)})(X_i^{(l)})$, which is the exact definition of constraints in Sec. 3. Finally, we need to show that no other competing source of information (via aspects containing parents or other interventions) breaks the constraints. Constraint (C3) requires the parents of (all aspects and in particular) $X_i^{(l^\star)}$ to be cut. By constraint (C2) no other intervention $do(X_i^{(l')} = x), l' \neq l$ can be placed on any of the aspects in $X_i^{(L)}$. Therefore, no other –potentially conflicting– signal influences $X_i^{(l')}$ and the values for all aspects are uniquely computed for the intervened case.

As such, we have shown the admissibility of variable configurations in DCM under constrains $\Phi$ for the intervened and unintervened case. ∎

## C  RELAXATION OF CONSTRAINT (C1)

Condition (C1) as described in Sec. 3 poses a rather strong constraint on the structure of the causal graph in order to guarantee consistency. Depending on the specific use-case the constraint might

be relaxed and replaced by $\arg\max_{x^\star} p(X_i^{(L)} = x^\star \mid \mathrm{pa}(X_i^{(L)}))$ given a (probability) measure $\mu$, that measures the violation of constraints $\phi_i^{(l,l')}$ with $\epsilon_i^{(l,l')} = \varphi_i^{-(l')} \circ X_i^{(l')} - \varphi_i^{-(l)} \circ X_i^{(l)}$ and $\mu(\otimes_{\{l,l' \in L_i, i=1\dots M\}} \epsilon_i^{(l,l')})$. This results in the most likely configuration (the one that violates constraints the least). This formulation might require solving for all aspect values under the implicit constraints $\phi^{(l)}$ and requires that a solution with non-zero probability always exists: $p(X_i^{(L)} = x^\star \mid \mathrm{pa}(X_i^{(L)})) > 0$.

## D  GREEDY DERIVATIVE MODEL SEARCH

In the following, we provide pseudo code for the `GreedyDerivativeModelSearch` in Algorithms 1-4 as described in the paper. We provide an implementation of the algorithm in the corresponding code repository: `https://anonymous.4open.science/r/derivativeCausalModels-E62B/`. Note, that the actual implementation aborts GDS in cases of no more edges being added or removed during `GDSForwards` or `GDSBackward`. Additionally, we cache the evaluation results of nodes given a specific set of parents. Both optimizations do not impact the correctness of the algorithm. We have spared them from the pseudo code to improve readability.

We make use of the following variables: $\mathbf{D}$ is the data, $N$ is the number of variables, Adj is the adjacency matrix of the to be discovered graph. Every edge is a pair $(i, j)$ where $j$ is a parent of $i$. edgeDelta measures the change in score of adding the edge $(i, j)$ to Adj, while edgeDelta$'$ measures the change when removing $(i, j)$ from Adj.

---

**Algorithm 1** Greedy Derivative Model Search

---

1: **procedure** GREEDYDERIVATIVEMODELSEARCH($\mathbf{D}$, numAspects)
2:     $\mathbf{D}' \leftarrow$ AugmentWithAspects($\mathbf{D}$)
3:     Adj $\leftarrow$ zeroMatrix($N \times$ numAspects, $N \times$ numAspects)
4:     **for** $i$ in $1..N$ **do**              $\triangleright$ Repeat phases N times, as suggested by Mian et al. (2021).
5:         Adj $\leftarrow$ GDSFoward(Adj, $\mathbf{D}'$)
6:         Adj $\leftarrow$ GDSBackward(Adj, $\mathbf{D}'$)
7:     **end for**
8:     **return** Adj
9: **end procedure**

---

**Algorithm 2** GDSForward

---

1: **procedure** GDSFORWARD(Adj, $\mathbf{D}$)
2:     bestScore $\leftarrow \infty$
3:     **while** True **do**
4:         edgeScores $\leftarrow [\,]$
5:         candidateEdges $\leftarrow$ allowedEdges(Adj)          $\triangleright$ Select all edges permitted by (C1)-(C3).
6:         **for all** $(i, j) \in$ candidateEdges **do**
7:             edgeScores $\leftarrow$ edgeDelta($i, j$, Adj, $\mathbf{D}$)  $\triangleright$ Compute score change when adding j $\rightarrow$ i.
8:         **end for**
9:         $i^\star, j^\star =$ candidateEdges[$\arg\min_i$ edgeScores]
10:         Adj[$i^\star, j^\star$] $\leftarrow 1$
11:         newScore $\leftarrow$ ScoreGraph(Adj)
12:         **if** newScore $\geq$ bestScore **then**
13:             Adj[$i^\star, j^\star$] $\leftarrow 0$ $\triangleright$ Adding the best edge did not improve the score; Revert and abort.
14:             **break**
15:         **end if**
16:         bestScore $\leftarrow$ newScore
17:     **end while**
18:     **return** Adj
19: **end procedure**

---

---

**Algorithm 3** GDSBackward

---

1: **procedure** GDSBACKWARD(Adj, $\mathbf{D}$)
2:     bestScore $\leftarrow \infty$
3:     **while** True **do**
4:         edgeScores $\leftarrow [\ ]$
5:         candidateEdges $\leftarrow$ where(Adj,1)            ▷ Collect all edges currently present in Adj.
6:         **for all** $(i, j) \in$ candidateEdges **do**
7:             edgeScores $\leftarrow$ edgeDelta$'(i, j, \text{Adj}, \mathbf{D})$
8:         **end for**
9:         $i^\star, j^\star = $ candidateEdges$[\arg\min_i \text{edgeScores}]$
10:        Adj$[i^\star, j^\star] \leftarrow 0$
11:        newScore $\leftarrow$ ScoreGraph(Adj)
12:        **if** newScore $\geq$ bestScore **then**
13:            Adj$[i^\star, j^\star] \leftarrow 1$ ▷ Removing the 'best' edge did not improve BIC; Revert and abort.
14:            **break**
15:        **end if**
16:        bestScore $\leftarrow$ newScore
17:     **end while**
18:     **return** Adj
19: **end procedure**

---

**Algorithm 4** ScoreNode

---

1: **procedure** SCORENODE($\mathbf{D}_{\text{node}}, \mathbf{D}_{\text{parents}}$)
2:     $w \leftarrow$ LASSO($\mathbf{D}_{\text{node}}, \mathbf{D}_{\text{parents}}, \alpha$)
3:     residuals$_A \leftarrow w \times \mathbf{D}_{\text{parents}} - \mathbf{D}_{\text{node}}$
4:     scores $= [\ ]$
5:     **for all** $p \in 1..P$ **do**
6:         $w_p \leftarrow$ fitPolynomial($p$, residuals$_A$)        ▷ Fit residuals with a polynomial of degree $p$.
7:         residuals$_B \leftarrow$ predictPolynomial(residuals$_A$, $w_p$) $-$ residuals$_A$
8:         $\sigma^2 \leftarrow$ variance(residuals$_B$)
9:         score $\leftarrow$ BIC($|\text{parents}| + p, \sigma^2$)
10:        scores $=$ append(scores, score)
11:     **end for**
12:     **return** $\arg\min_i(\text{scores})$
13: **end procedure**

---

Note that after LASSO regression in Line 2 parents might be removed if their weights are found to be zero (as described in Sec. 4.1). We have ommited this logic in the pseudo code to improve readability.

### D.1 GREEDY DERIVATIVE MODEL SEARCH EVALUATION AND DATA SET GENERATION

In the following we describe the setup of our synthetic time series data set and present further evaluations altering the number of nodes and density of the graph.

**Data Set Generation.** A coarsened graph is created by sampling a given percentage of entries from a lower triangular matrix. One derivative and one integral aspect per variable are created. For every edge in the coarsened graph a random subset of the aspects is chosen as parents. Edge weights are uniformly sampled from the range $\pm(0.1, 1.0)$. Starting values at $t = 0$ of all aspects are uniformly sampled in the range $[-2, 2]$. Values of root variables are uniformly sampled in the range $[0.1, 1]$ for all timesteps. For every endogenous variable normal gaussian noise $\mathcal{N}(0, 1)$ is added. For every graph 10 different starting configurations are sampled and a time series of 50 steps is computed via finite differences (derivative aspects) and summation (integral aspects). Every configuration altering the number of nodes or edge percentage is created are evaluated from 20 different seeds which alter random sampling in all above processes.

**Evaluation.** The following table contains the results of the experiment as shown in the main paper, with 6 nodes (n=6) and 50% chance of an edge in the coarsened graph to exist (e=50%). Additionally, we include SHD over the coarsened graph in this and show results for fixing all other parameters and varying to number of nodes and edge percentage, respectively.

| GDS; n=6; e=50% | Precision | Recall | SID | SHD |
|---|---|---|---|---|
| $\alpha = 0.01$ | 63.47% (20.44) | 60.21% (22.16) | 4.95 (3.82) | 4.40 (1.65) |
| $\alpha = 0.1$ | 62.87% (20.34) | 59.50% (22.48) | 5.00 (3.84) | 4.50 (1.65) |
| $\alpha = 1.0$ | 61.03% (24.78) | 49.19% (21.07) | 6.10 (3.98) | 4.70 (1.58) |
| $\alpha = 10.0$ | 62.66% (30.54) | 28.88% (15.61) | 8.85 (3.75) | 5.30 (1.38) |
| $\alpha = 100.0$ | 61.16% (35.28) | 26.21% (17.84) | 8.80 (3.85) | 5.05 (1.20) |
| linear regr. | 60.74% (19.92) | 60.59% (24.73) | 4.85 (3.90) | 4.45 (1.59) |
| no poly. fit. | 25.68% (15.08) | 38.40% (24.56) | 9.80 (4.01) | 9.75 (2.42) |

| GDS; e=50%; $\alpha$=1.0 | Precision | Recall | SID | SHD |
|---|---|---|---|---|
| $n = 4$ | 64.16% (42.58) | 46.49% (33.14) | 1.65 (1.76) | 1.55 (0.73) |
| $n = 6$ | 61.03% (24.78) | 49.19% (21.07) | 6.10 (3.98) | 4.70 (1.58) |
| $n = 8$ | 52.92% (14.72) | 49.00% (15.42) | 16.25 (6.56) | 10.00 (2.36) |
| $n = 12$ | 46.46% (13.00) | 45.93% (10.45) | 50.90 (14.46) | 25.90 (5.00) |
| $n = 16$ | 39.94% (7.61) | 38.03% (8.12) | 132.65 (24.58) | 51.25 (6.36) |

| GDS; n=6; $\alpha$=1.0 | Precision | Recall | SID | SHD |
|---|---|---|---|---|
| $e = 20\%$ | 60.00% (46.36) | 43.33% (38.87) | 1.35 (1.42) | 1.50 (1.02) |
| $e = 30\%$ | 59.16% (40.98) | 35.83% (28.39) | 2.50 (2.31) | 2.35 (1.15) |
| $e = 40\%$ | 57.70% (32.94) | 50.24% (30.41) | 3.80 (3.02) | 3.65 (1.27) |
| $e = 50\%$ | 61.03% (24.78) | 49.19% (21.07) | 6.10 (3.98) | 4.70 (1.58) |
| $e = 60\%$ | 62.39% (22.59) | 52.13% (20.95) | 8.05 (3.47) | 5.35 (1.35) |
| $e = 70\%$ | 58.27% (19.66) | 51.51% (15.83) | 8.05 (3.48) | 6.00 (1.67) |
| $e = 80\%$ | 62.80% (20.04) | 47.58% (15.14) | 12.10 (3.88) | 7.60 (2.31) |

**Evaluation with GES.** While GES is bound to discover CPDAGs (Meek, 2013) it could be, that GES might still discover valid DCM from data. To test this hypothesis, we apply GES in our standard setting (n=6,e=50%) on our data with all derived aspects generated, to obtain a CPDAG. We then generate all possible DAG realizations of the CPDAG –which means that we generate DAGs containing all possible combinations of directed edges for all undirected edges– and test if they are valid DCMs. To make for a fair comparison, we use our BIC with polynomial fitting as the scoring function. We find that over all 20 seeds GES is not able to recover a graph that aligns with the DCM constraints.

Nonetheless, GES is able to recover relations between individual aspects. We therefore provide all aspect variables individually, and let GES identify the causal relations between them. After discovery, the individual aspects are then grouped together to form the coarsened graph. Given that GES might have predicted contradicting edge directions among the grouped aspects from or to other variables, we decide the edge direction by a majority vote. If there is an equal number of edges in either direction, we give the benefit of doubt and direct the coarsened edge using the correct direction of the ground truth graph.

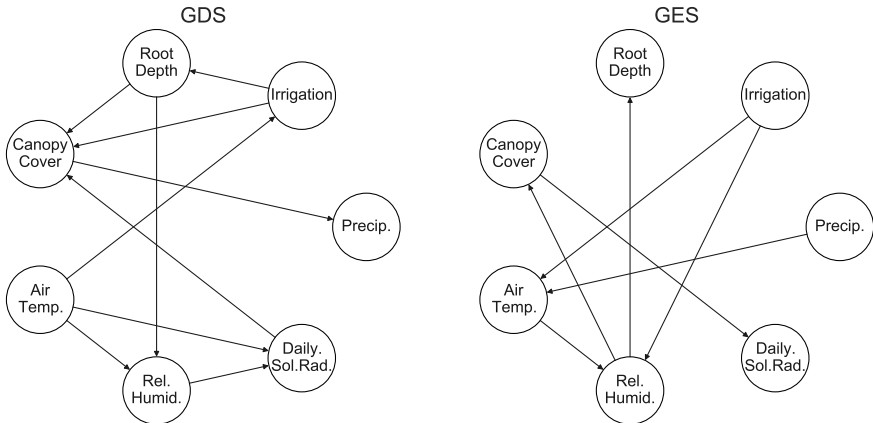

Figure 5: **Causal graph discovery on the USDA-ARS Colorado Maize Water Productivity Dataset.** The real-world data set resembles the setup of the motivational example of the introduction. Discretely occurring effects, such as precipitation and irrigation are mixed with continuously changing variables such as the canopy cover. Results are show for GDS (left) and GES (right). We find that our GDS algorithm more closely predicts the structure predicted in our idealized motivational example.

**Compute Time.** All experiments where run on an AMD Threadripper 1900X 8x 3.80GHz. Full evaluation of all experiments took 29m 52s. A run with n=6, e=50%, $\alpha$=1.0 over 20 seeds took 46 seconds, while evaluating 20 seeds over graphs with 16 nodes took 10 minutes.

### D.1.1 USDA-ARS COLORADO MAIZE WATER PRODUCTIVITY DATASET

We present an evaluation on the publicly available "USDA-ARS Colorado Maize Water Productivity Dataset" (https://catalog.data.gov/dataset/usda-ars-colorado-maize-water-productivity-dataset-2008-2011-5460b) which is similar to the idealized setup of our motivational example. We predict the causal graph for the 2011 subset over the following variables: precipitation, irrigation ('water sources'); root depth and canopy cover ('plant metrics'); avg. air temperature, relative air humidity and daily solar radiation ('weather conditions'). We fuse the provided weather data with the plant records during their growth phase and apply our algorithm with 2 integral and derivative aspects. As GES performs worse when provided with additional aspect variables, we present its direct evaluation on the pure set of variables. Graph predictions are provided in Fig. 5.

Note that the irrigation and precipitation variables are mostly sparse ('zero-valued'), such that water is only provided to the plant every few days. Apart from other 'standard' causal relations, our algorithm is able to identify many edges expected by common sense. Specifically GDS identifies 'Irrigation → Root Depth' and 'Irrigation → Canopy Cover', resembling the predicted 'Watering → Plant Growth' *integral relation*, while GES is unable to do so. We find that also the edges 'Canopy Cover → Precipitation', 'Root Depth → Relative Air Humidity' and all in-going 'Daily Solar Radiation' edges are discovered but are pointing in the wrong direction.

When supplied with the exact same data of two integral and derivative aspects GES is unable to identify any edges at all. Lowering augmentation to a single integral and derivative aspect, GES finds 'Irrigation ↔ Precipitation'. When providing the data with no additional augmentation, GES identifies the graph shown in the figure. In detail, 'Irrigation, Air Temperature → Relative Air Humidity' and 'Precipitation → Air Temperature' might be considered correct. However, none of the previously identified integral relations are identified.

# E    EXAMPLE ON GAME DYNAMICS

Below we show the potential value sequence of the "Capturing Game Dynamics" example as shown in Fig. 4 (Sec. 4.2).

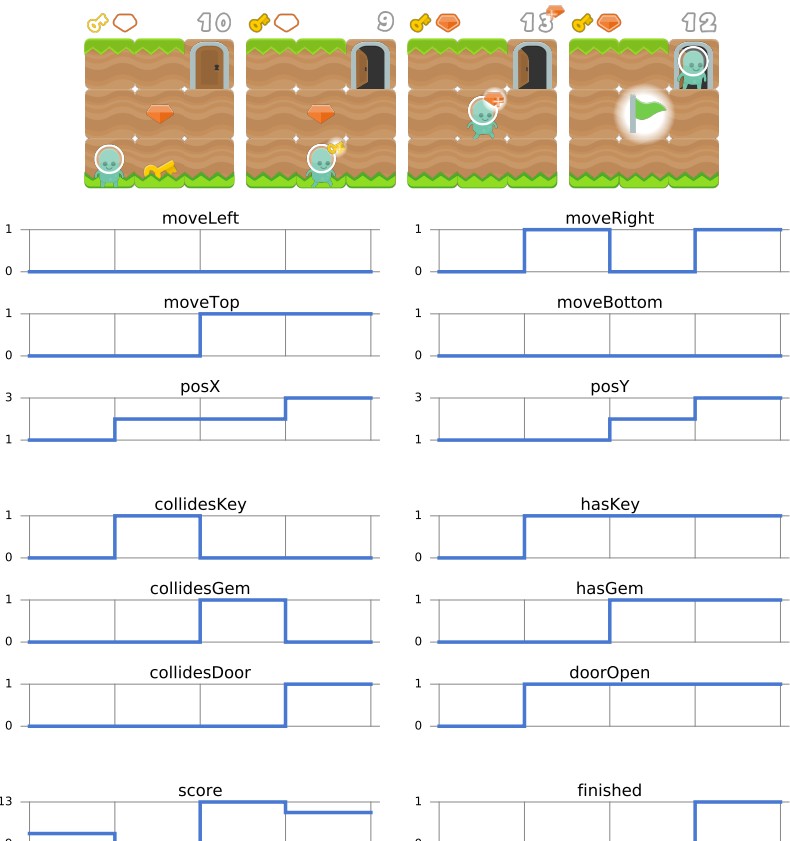

Since we observe a settings without noise, many of the relations can only be observed as correlations. However, no direction can be determined for one-to-one relations due to symmetries in the data. This issue could be resolved by placing additional assumptions on our data. For example, one could use typing information (Brouillard et al., 2022) or language models (Zečević et al., 2023; Long et al., 2023) to figure out directions via external information. Alternatively, player inputs could be considered interventions on the system, such that all edges (an recursively their relations) can be directed 'away' from the intervened variables.

**Inference.** DCM might be used to reason backwards in time by leveraging derivative aspects. In this particular setting one can for example leverage the presence of `MoveLeft/Right` as derivatives of `positionX`, to infer the position of the player at the previous time step $t-1$, such that $positionX_{t-1} = positionX_t - moveLeft_t + moveRight_t$, and similar for $hasKey_{t-1} = hasKey_t + collidesKey_t$... By having access to further higher-order derivatives or by placing additional smoothness assumptions on the derivatives in continuous settings, one might be able to extend such inferences beyond a single time step and reason backwards over longer periods of time.

**Asset Sources**: Individual assets used in Figure 4 where taken from `https://www.kenney.nl/assets/platformer-pack-redux` licensed under Creative Commons CC0.

