# OpenReview forum: "Derivative Causal Models: Modeling Causality at Mixed Scales of Observation"
_ICLR.cc/2025/Conference — ICLR 2025 Conference Withdrawn Submission_

### Official Review · Reviewer_ewB2 · 2024-11-01

**Soundness:** 4
**Presentation:** 4
**Contribution:** 2
**Rating:** 5
**Confidence:** 3

**Summary:**

The paper introduces Derivative Causal Models (DCM) as causal models for explicitly modeling mixed-scale data, such as combining discrete events and continuous measurements. Traditional methods, such as Structural Causal Models (SCM), assume smooth and continuous variations in cause-effect relationships. DCM instead uses mathematical transformations to derive complex relationships into simpler linear parts with constraints. The manuscript also introduces one realization of a DCM, called Differential Causal Models (dCM), where transformations are restricted to integrals and derivatives. Another contribution of this paper is the Greedy Derivative Model Search (GDS) algorithm - an extension of Greedy Equivalence Search (GES) - which learns a causal graph structure by using the Bayesian Information Criterion and LASSO regularization in the context of the constraints introduced by a DCM. The paper presents experimental results on synthetic and modeling analysis on real-world data capturing cause-effect interactions in a gaming scenario.

**Strengths:**

* Strong theoretical foundation: The manuscript has an excellent presentation and describes its formal models with rigorous writing. For instance, the paper presents DCM by introducing theoretical constraints in Section 3, showing how admissibility of an SCM in Theorem 3.1, and finally defining a DCM in Definition 2. Section 3.1 similarly introduces dCM as a set of convenient transformations for DCM.
* Proposed algorithm GDS for causal discovery, based on GES, is capable of handling transformation constraints while yielding a valid DCM.

**Weaknesses:**

* Interpretability of aspects in transformed variables: it is unclear from the manuscript how transformations affect the original understanding of a variable.
* Model expressiveness: the manuscript acknowledges that the introduced constraints C1, C2, and C3 may limit the expressiveness of the models. However, no further practical discussion on this constraint is provided in the paper. For instance, it would be valid to understand the practical value of the trade-off between expressiveness and admissibility.
* Related works discussion: The relationship with Constraint Causal Models (CCM) (Blom et al., 2020) could be better explored by improving CCM background information and examples. Although the paper discusses how CCM constraints are expanded for transformed variables in the section “Admissibility and Interventions,” the following points on solving constraints, effect on interventions, and admissibility are convoluted.
* Experimental results aren’t convincing: the manuscript presents encouraging results for GDS but does not clearly demonstrate the modeling benefits of a DCM, the manuscript's main contribution. Section 4.2 briefly discusses how dCM can be used to model game dynamics. However,  the paper could benefit from longer and more insightful experiments with the dCM modeling capabilities.

**Questions:**

* Could you further discuss the challenges of the Greedy Derivative Model Search (GDS) implementation? Please include comments on complexity, magic numbers insights (such as repeating backward-forward 5 times), time/cost to convergence, etc.
* What are some insights from the results in Figure 3?

---

### Official Review · Reviewer_BUxK · 2024-11-02

**Soundness:** 2
**Presentation:** 2
**Contribution:** 3
**Rating:** 6
**Confidence:** 3

**Summary:**

The paper tackles the problem of constructing causal models on variables observed at different "*scales*".  To this end, the authors assume that each variable $X\in\set{X}$ can undergo one or more bijective transformations $\varphi_X^l$, where $l\in\set{L}$ is the set of total transformations. The authors then propose to consider a causal graphs as a connection between *aspects*, i.e., realizations of variables after one of these transformations. They then introduce a specialization of DCMs, called $\partial$CMs, where the transformations consist of the operations of integration and derivation of the variable and propose an algorithm to fit DCMs/$\partial$CMs from data, which they apply to both simulated and real-world data.

**Strengths:**

The paper presents interesting and challenging points on the design of causal models, and it's in my opinion worth presenting and discussion at the venue. The problem of mixing observations at different scales is pressing in real-world applications, and DCMs constitute an interesting formalization of the problem.

**Weaknesses:**

I found the definition of *scale* quite confusing as "the nature of certain variables to exhibit different qualities in observation" is either too vague, if we assume common-sense understanding of "nature" and "quality", or too specific, if the authors intend them as philosophical terms. In both cases, it would help if the authors were to more clearly introduce their definition. Notably, their definition of scale might contrast with the more intuitive meaning, at least for the ICLR community, of the *scale* of a variable as related to the variance of a random variable. Similarly, Theorem 3.1 is non sufficiently formalized, for which I'm leaving further comments in the questions section. In general, the presentation of the paper should be more precise and formal.

I have some doubts on Section 3.2 on the compact representation of DCMs. While the operation reduces the number of nodes visually, it does not compact the graph and resulting operations still need to process the "hidden" nodes. While leading to arguably nicer *drawings*, the representation has no practical impact on the applications. In comparison, this seems like taking the SCM in Figure 1 — the *classical* one, and drawing an edge $W \to S$ without the latent intermediate variables.

The methodology for the discovery of DCMs is not sufficiently explained in the main paper, as the algorithm is relegated to the appendix.

**Questions:**

- In your motivating example on watering plants, you claim that lagged models could be used but that the resulting representation would be more convoluted (l. 037). This seems like an important point to justify your proposal: could you elaborate on why the representation would be more convoluted?
- Still related to the previous question, I'm not sure about the claim on Line 160 on the independence of the watering event and the flower size. Intuitively, it seems to me that you *could* relate the growth in flower size to watering in a lagged causal model. I would expect $W_t$ and $F_{t+\varepsilon}$ to be marginally dependent given a temporal dataset for a $\varepsilon$ representing the *necessary* lag. Why would this be problematic compared to DCMs?
- Whenever you compose functions with variables, as in $\phi_i^{(l)}\circ X_i$, are you implicitly referring to the structural equation of that variable? You defined $\mathbf{F}$ as the set of structural equations, but then they are never used in the paper.
- The statement in Theorem 3.1 should be formalized better. What's an evaluation? An admissible variable configuration? What do you mean by "with respect to"? Does it apply to SCMs or DCMs?
- How is the semantic grouping performed in the experiment on game dynamics in Section 4.2? Also, is the method applied on a tabular dataset or directly on the visual frames?

---

### Official Review · Reviewer_bdLa · 2024-11-03

**Soundness:** 3
**Presentation:** 2
**Contribution:** 2
**Rating:** 5
**Confidence:** 2

**Summary:**

The paper studies a causal representation for mixed scales of observables in real-world systems.  Author leverage transformations that derive different scales of observables, to decompose relations and allow for compact causal representations, called Derivative Causal Models (DCM). A greedy score-based algorithm is further proposed to discover differential causal models from data.

**Strengths:**

- The paper tackles a practically meaningful and important scenario where variables have different scales.

- The proposed model seems reasonable.

**Weaknesses:**

- Some key concepts/background are not clear, and clarity can be much improved.

- No identifiability analysis for the proposed casual discovery method.

**Questions:**

The paper tackles a practically meaningful and important scenario. However, it is quite struggling to go through and I can only give a sensible review.

The paper is not self-contained in several places:

- regarding the definition of "scale": the "scale" is a key concept in this paper, and it is only explained till Section 3: "By scale we mean the nature of certain variables to exhibit different qualities in observation and thus to be measured by different means and frequencies." Even with this explanation, the concept "right scale", which is used several times, has no clear definition and meaning. To be clear, how can we know a variable is measured in the "right scale", by a second, an hour, a day or a week for the example Watering example? Please clarify.
- CCM and CMC are interesting models, but are (at least for me) of a small research area. So please introduce more background.
- some paragraphs are too long and can be separated to improve readability, e.g., paragraphs on page 4.

Regarding identifiability:

- since a GES type algorithm is developed, I would like to ask what is the identifiability condition here. An identifiability  and identification analysis is important to learning causal structures from the data.

---

### Official Review · Reviewer_EZtM · 2024-11-03

**Soundness:** 1
**Presentation:** 2
**Contribution:** 2
**Rating:** 3
**Confidence:** 4

**Summary:**

Summery:
The paper introduces the Derivative Causal Models (DCM) and aims at addressing the challenges of finding causal relations from observations at different time scales. The paper utilized derivatives and integrals, along with reformulation of causal models, under three key assumption and constraint to simplify non linear relations to linear relations. The paper formalizes DCM and presents a specialized algorithm, Greedy Derivative Model Search (GDS), for identifying these models within data. This new approach is a more compact way to represent multi-scale causal systems.

**Strengths:**

- The Greedy Derivative Model Search (GDS) algorithm is designed to identify DCMs automatically from data, which in presence of adequate theoretical grounds could be a helpful tool.

- The Greedy Derivative Model Search (GDS) algorithm is well presented and explained in the appendix.

- The code for the paper is provided, suggesting that is is easy to reproduce the results.

**Weaknesses:**

- The integration of transformation and causal variables is not rigorously grounded. In fact this notion seems to have some fatal flaws. For example, the authors don’t discuss how their frameworks handles transformations such as $f(x)$, for which $f^{-1}(x)$ does not exist. Having based the core of the paper on this notion, I expect that these mathematical details be extensively proved, validated and grounded.

- The paper requires three key constraints for the method to be complete and correct. However, there is no discussion of how common or extreme these constraints are. It appears that these constraints rule out a big portion of practical use cases.

- There is no discussion of how the premises of the paper will hold if the assumption( which there are many) are violated.

- The proof for Theorem 3.1 is not an actual proof, just restating constraints 1 to 3.

- The paper could benefit from more useful illustration and figures. Aside from Figure 2, the rest of the figures do not have any additional information and don’t help the reader understand the concepts of the paper.

- Their result section is extremely small and is limited a single table, comparing to only two other methods.

- Many related works are not cited or mentioned. For example these papers all address causal inference from different time scales, but there is no mention of them in this paper.

	* Learning Multiscale Non-stationary Causal Structures (by Gabriele D'Acunto, Gianmarco De Francisci Morales, Paolo Bajardi, Francesco Bonchi)
	* Causal Learning through Deliberate Undersampling (by Kseniya Solovyeva, David Danks, Mohammadsajad Abavisani, Sergey Plis)
	* Causal Discovery from Temporally Aggregated Time Series (by Mingming Gong , Kun Zhang , Bernhard Schölkopf , Clark Glymour , Dacheng Tao)

To name a few

- The introduction section should be a valid text by it self. In this paper it starts with referring to the abstract. It is a good practice to present the problem the paper is addressing clearly in the introduction section.

- Some abbreviations, like SCM, are used multiple times without first defining what they stand for.

**Questions:**

- How common are the assumptions of the paper?
- What happens if they are violated?
- How is your method compared to other published works?

Recommendation: The paper needs a solid results section with more tests and comparisons. Further investigation is needed to confirm its robustness and broader applicability.

---

### Author Response · Authors · 2024-11-22

Dear reviewers,
thank you for your time and effort in reviewing our paper. We still believe that our presented algorithm can be relevant for a broad range of real-world applications, in cases where aspects of variables relate on different scales (consider the Maize Watering example) and for which existing algorithms do not yet pose a solution. However, testing independence across different aspects, which might yield stronger identifiability results, poses a major challenge that we are not aware of to have been overcome yet. Still, we acknowledge the mentioned weaknesses in presentation and will take our time and your comments seriously to revise the paper. Thank you once again.

Best Regards,
Authors

---

### Note · Authors · 2024-11-22

I have read and agree with the venue's withdrawal policy on behalf of myself and my co-authors.